# MAGEA4 Coated Extracellular Vesicles Are Stable and Can Be Assembled In Vitro

**DOI:** 10.3390/ijms22105208

**Published:** 2021-05-14

**Authors:** Olavi Reinsalu, Anneli Samel, Elen Niemeister, Reet Kurg

**Affiliations:** Institute of Technology, University of Tartu, 50411 Tartu, Estonia; olavi.reinsalu@ut.ee (O.R.); anneli.samel@ut.ee (A.S.); elen.niemeister@gmail.com (E.N.)

**Keywords:** extracellular vesicles, cancer-testis antigen, MAGEA4, recombinant protein

## Abstract

Extracellular vesicles (EVs) are valued candidates for the development of new tools for medical applications. Vesicles carrying melanoma-associated antigen A (MAGEA) proteins, a subfamily of cancer-testis antigens, are particularly promising tools in the fight against cancer. Here, we have studied the biophysical and chemical properties of MAGEA4-EVs and show that they are stable under common storage conditions such as keeping at +4 °C and −80 °C for at least 3 weeks after purification. The MAGEA4-EVs can be freeze-thawed two times without losing MAGEA4 in detectable quantities. The attachment of MAGEA4 to the surface of EVs cannot be disrupted by high salt concentrations or chelators, but the vesicles are sensitive to high pH. The MAGEA4 protein can bind to the surface of EVs in vitro, using robust passive incubation. In addition, EVs can be loaded with recombinant proteins fused to the MAGEA4 open reading frame within the cells and also in vitro. The high stability of MAGEA4-EVs ensures their potential for the development of EV-based anti-cancer applications.

## 1. Introduction

The scientific interest in extracellular vesicles (EVs) has grown exponentially during the last decade. A vast amount of research has been conducted to unravel the cellular biology of EVs and to find and develop new approaches for using the potential of EVs in a variety of applications. “Extracellular vesicle” is an umbrella term for nano-sized particles with a phospholipid bilayer membrane derived from the cells. As EVs are released into extracellular space essentially by all cells, they can be found from all body fluids, e.g., blood, urine, saliva, semen, synovial fluid, and others [1,2,3,4,5]. There are three major types of EVs: exosomes, microvesicles, and apoptotic bodies. The most significant difference between these types is the biogenesis pathway of the vesicles. Other characteristics such as size, morphology, and protein content are not sufficient for their clear distinction [6], hence the use of the umbrella term. 

EVs were discovered in the mid-20th century by pelleting normal blood plasma [7]. For a long time, the biological importance of the vesicles remained unclear and was mostly considered as a residue from platelets [8]. A few decades ago it was discovered that EVs were not waste compartments but rather carriers of bioactive cargo, such as nucleic acids or proteins, which is packed inside the protective lipid bilayer and can be delivered to other cells and alter the intracellular processes of the recipient cells [9,10,11,12]. Driven by these discoveries, EVs are now recognized as a vital part of intercellular communication, having an important role in physiological homeostatic and pathological processes [13,14,15]. For example, EVs have been shown to participate in angiogenesis, coagulation, tissue repair and regeneration, regulation of immune response, and inflammation [6,16,17,18,19]. On the other hand, EVs are associated with the formation of cardiovascular or neurological diseases [20,21,22,23,24] and are also known to be involved in cancer development and metastasis [25,26,27,28,29].

EVs are also recognized as potential candidates for medical applications, including diagnostic and therapeutic approaches [30,31,32]. In the context of cancer diagnosis and prognosis, EVs have shown to be a great source for cancer biomarkers as they can be obtained by minimally invasive liquid biopsies [33,34] and because EVs derived from cancer tissue reflect the macromolecular composition of cancer cells, including specific sets of RNA and proteins [35,36,37]. Cancer-testis antigens are a large group of proteins considered to be valued candidates for cancer biomarkers as these proteins are expressed in a variety of tumors but not in any healthy tissue except for germlines and placenta [38,39]. Among cancer-testis antigens, melanoma associated antigen A (MAGEA) is a subfamily of proteins that are found to be expressed mostly in cancers that have formed a malignant phenotype, have a high mutation burden, and are invasive and metastatic [40,41,42]. 

The MAGEA4 protein, a member of the MAGEA protein subfamily, is highly expressed by numerous tumors of different histological types, including urothelial carcinoma, oral squamous cell carcinoma, lung cancer, ovarian neoplasm, and others [43,44,45,46]. It has been shown that cancer patients can have a naturally occurring antibody response against MAGEA4 [47]. With the use of MAGEA4 in immunotherapeutic techniques, the immune response could be stimulated even further, causing retreating effects of cancer [48,49,50]. Attempts of clinically approved anti-cancer therapeutic approaches involving MAGEA4 have been made [50,51,52]. All these efforts describe MAGEA4 as a highly valued candidate for developing cancer-targeting immunotherapies. Previously, we have reported that MAGEA4 is metabolically incorporated into EVs and retrovirus Gag protein-induced virus-like particles (VLP) in cancer cells as well as in cells ectopically expressing the protein where the MAGEA4 protein is exposed on the surface of the vesicles [53,54]. This phenomenon is highly intriguing, considering that MAGEA4 is a soluble cytoplasmic protein in cells.

In the current work, we have studied the biochemical properties of MAGEA4-EVs in more detail. EVs with different cargo might behave differently in similar conditions [55], therefore, it is important to study the properties of the EVs that specifically carry the observed cargo. Here, we examined the biophysical and -chemical stability of EVs carrying MAGEA4 under common storage conditions and their resistance to freeze-thaw cycles. In order to further elucidate the attachment of MAGEA4 to the outer surface of the EVs, we also investigated the hypothesis by which MAGEA4 is bound to the surface of the vesicles as a peripheral membrane protein. Furthermore, new features of MAGEA4 were explored to obtain a better understanding of its potential for downstream EV-based applications.

## 2. Results

### 2.1. MAGEA4-EVs Are Stable under Common Storage Conditions

MAGEA4-EVs used in this study were isolated and purified from cell culture media as described previously [54]. Briefly, for the production of MAGEA4-EVs, mouse COP5-EBNA fibroblast cells were transfected with a MAGEA4-expressing plasmid, and the isolation and purification of EVs were performed 72 h post-transfection using differential ultracentrifugation. The characterization of MAGEA4-EVs is shown in Appendix A. In order to assess the stability of MAGEA4-EVs under common storage conditions, purified MAGEA4-EVs were divided into two groups of equal aliquots and were stored at −80 °C or 4 °C for up to 3 weeks. After every 7 days a sample was taken from both groups and analyzed by Western blot, flow cytometry and nanoparticle tracking analysis (NTA). 

The Western blot analysis did not show any remarkable difference in the amount of MAGEA4 protein in EV samples stored either at 4 °C or −80 °C for 3 weeks (Figure 1A). In order to analyze whether the MAGEA4 protein is still attached to the surface of EVs, we performed a flow cytometry analysis with antibodies recognizing the MAGEA4 protein. As shown on Figure 1B, the mean fluorescence intensities (MFI) of samples from different time points kept at different temperatures did not show any statistically significant difference. Furthermore, the fluorescent intensity profiles were almost identical to each other, showing high homogeneity of samples (Figure 1C). NTA analysis also confirmed that there were no statistically significant changes in the number of particles (Figure 1D) or mean diameter (Figure 1E) between treated samples at different time points or compared to controls in both groups. This suggests that MAGEA4-EVs are very stable under common storage conditions for at least up to 3 weeks after purification.

### 2.2. MAGEA4-EVs Are Stable to Freeze-Thaw Cycles

For the assessment of freezing resistance of the MAGEA4-EVs, the vesicles were subjected up to three cycles of freezing and thawing. A single cycle comprised of 1 h of freezing at −20 °C and 20 min of thawing at room temperature. After each cycle, an aliquot was analyzed by Western blotting, flow cytometry, and NTA analysis. Again, MAGEA4 specific fluorescence profiles in flow cytometer analysis (Figure 2A) were very similar, but this time having slightly greater variance in homogeneity. Here, the homogeneity of EVs was increased with each freeze-thaw cycle. The amount of MAGEA4 on the surface of EVs (Figure 2B) and in EV lysate (Figure 2C) started to decrease only after the third freeze-thaw cycle (*p*-value 0.025), indicating a slight loss of EVs carrying MAGEA4. Although it was statistically not significant, we observed a slight increase in EV numbers after the first freeze and thaw cycle, following a decrease with every next cycle (Figure 2D). Variations in the size distribution of EVs after different treatments are shown in Appendix A. A slight but statistically significant growth in EV diameter after the first (*p*-value 0.03) and second cycle (*p*-value 0.026) was also detected (Figure 2E). This could be explained by the disruption of aggregates after the first cycle releasing more EVs and thus increasing the homogeneity. An alternative explanation is that during the first cycle faulty and immature EVs were degraded and remaining EVs are more homogenous. In summary, this experiment showed that MAGEA4-EVs could be freeze-thawed at least two times without any loss of MAGEA4 cargo. 

### 2.3. MAGEA4-EVs Are Resistant to the Treatment with High Salt

To test the hypothesis that MAGEA4 associates with EVs as a peripheral membrane protein, we treated MAGEA4-EVs with chemicals that are commonly used for extracting peripheral membrane proteins while keeping the membrane and transmembrane proteins intact [56]. MAGEA4-EVs were divided into aliquots containing equal amounts of vesicles, treated with different solutions for 1 h at room temperature and then washed with PBS using ultracentrifugation (Figure 3A). Reagents were dissolved in PBS, while MAGEA4-EVs treated with PBS alone was used as a positive control. In the flow cytometer analysis, latex-sulfate beads without vesicles were used as a negative control.

First, MAGEA4-EVs were treated with 1 M NaCl and 0.33 M MgCl_2_, which are high ionic strength solutions with different chaotropicity, the latter being more chaotropic [57] and containing metallic ions with discrete valency. In order to have equal ionic strength of the solutions, NaCl solution used was three times more concentrated than MgCl_2_. As shown in Figure 3B,C, the amount of MAGEA4 on the surface of EVs was slightly different after treatment of high ionic strength salt solutions. Surprisingly, it was increased rather than decreased compared to the control sample (Figure 3B,C). The MFIs of MAGEA4-EVs in treated samples were higher with *p*-values of 0.0377 for NaCl and 0.0199 for MgCl_2_ (Figure 3D). There were no significant changes in the number nor diameter of the EVs in the salt-treated samples compared to the control, suggesting that MAGEA4-vesicles withstand high osmotic pressure very well (Figure 3E,F).

Second, chelators, EDTA and EGTA, were used to deprive the surface of the EVs from bi-valent metallic ions, including Ca^+2^ and Mg^+2^, which are essential for metal ion-dependent adhesion of proteins. The treatment with chelators (EDTA and EGTA) did not show any statistically significant difference compared to the control sample (Figure 3C,D).

Third, to assess the effect of high pH on MAGEA4 adhesion, NaOH with a pH of 11.5 was used. MAGEA4-EVs were also treated with Triton X-100 in order to create membrane disrupting conditions. Treatments of MAGEA4-EVs with 0.02% of non-ionic detergent Triton X-100 and high pH had a negative effect on the presence of MAGEA4 on vesicles. The fluorescence profiles showed a much more heterogeneous population of EVs (Figure 3B) while having lower fluorescence intensities. Although Western blot results showed a similar or slightly diminished amount of MAGEA4 (Figure 3C) compared to the control, the MFIs had clearly decreased (Figure 3D). The MFI *p*-values for samples treated with Triton X-100 and NaOH were 0.0282 and 0.0054, respectively, when compared to the control sample. Treatment with detergent or a high pH had a destructive effect on MAGEA4-EVs as the number of the vesicles had greatly diminished (*p*-values 0.0001 and 0.0006, respectively) (Figure 3E). A similar effect was observed after treatment with KOH at pH 11.5 (Appendix A). The amount of detected MAGEA4 was higher in Triton X-100 treated samples compared to NaOH treated samples (Figure 3C). Interestingly, the material treated with detergent contained particles with a slightly bigger diameter (*p*-value 0.0266) (Figure 3F) and had a marginal shift in the size profile compared to the control (Figure 3G) while other samples remained similar. We suggest that the detected increase in size might be due to aggregation of vesicle debris, where still some MAGEA4 protein is attached.

### 2.4. Purified MAGEA4 Protein Associates with Extracellular Vesicles In Vitro

To analyze the MAGEA4 protein attachment to vesicles in vitro, the MAGEA4 protein was expressed and purified from bacteria and mixed with EVs isolated from cell culture media of COP5-EBNA cells. In this case, EVs were purified from growing cells without any manipulation. For comparison, murine leukemia virus (MLV) Gag-induced virus-like particles (VLPs) were used [53]. For the production of VLPs, COP5-EBNA cells were transfected with MLV Gag expressing vector following the isolation of VLP from the cell culture medium 72 h post-transfection. The retrovirus Gag protein induces the budding of vesicles with very similar biochemical characteristics as natural EVs while having a better yield [53,58]. The purified MAGEA4 protein and EVs/VLPs were mixed and incubated in PBS for 1 h at room temperature, followed by separation using ultracentrifugation and size-exclusion chromatography (SEC) (Figure 4A).

As shown in Figure 4B, the MAGEA4 protein was detected on the vesicles after 1 h of incubation on the bench and ultracentrifugation through PBS. There was no difference between EVs and VLPs, in both cases, MAGEA4’s association with vesicles was easily detected by Western blot analysis (Figure 4B). Flow cytometry analysis confirmed that the MAGEA4 protein was efficiently attached to the surface of EVs as well as VLPs (Figure 4C). It is possible that MAGEA4 is attached to the vesicles during ultracentrifugation due to very high centrifugal force used during centrifugation. To exclude this, the VLPs were purified by size exclusion chromatography after in vitro incubation. Again, the MAGEA4 protein and VLPs were incubated for 1 h at room temperature and then loaded onto the column. As shown in Figure 4D, the MAGEA4 protein was detected in fractions 7–8 as well as in fractions 15–19. The MLV Gag protein, used as a marker of VLPs, was also in fraction 7–9 and the same appeared for TSG101, although the latter was weakly detectable (Figure 4D). Most of the MAGEA protein was fractionated into fractions 16–19 apart from VLPs, meaning that most of it was not bound to the vesicles. Nevertheless, there was a considerable amount of MAGEA4 in the vesicle fractions separated from soluble protein, suggesting that MAGEA4 has the ability to bind to the EVs without going through the biogenesis pathways of the vesicles in the cells of origin.

### 2.5. MAGEA4 Can Be Used for Decorating EVs with Recombinant Proteins

To further uncover the potential of MAGEA4 for targeting proteins to the vesicles, we used EGFP as a marker protein. COP5-EBNA cells were transfected with vectors expressing MAGEA4-EGFP chimeric protein or EGFP alone, and EVs were isolated from the cell culture media 72 h post-transfection. The expression of both proteins in cells was confirmed by Western blot (Figure 5A) and flow cytometry analysis (Figure 5B). Analysis of EVs showed that the MAGEA4-EGFP protein was incorporated into vesicles while EGFP was not (Figure 5A). Flow cytometry analysis measuring the intensity of EGFP fluorescence confirmed that the chimeric MAGEA4-EGFP was incorporated into vesicles and EGFP was not, as the fluorescence profile of EGFP closely matched the control sample (Figure 5C). Further staining with MAGEA4 antibodies showed that MAGEA4 was attached to the outer surface of EVs where it localizes together with EGFP as they were expressed as a fusion protein (Figure 5D). This suggests that MAGEA4 is capable of assigning cargo to the EVs in case they are fused together.

Finally, EVs isolated from non-transfected COP5-EBNA culture media were incubated with purified prokaryotic EGFP and MAGEA4-EGFP proteins and washed through ultracentrifugation as described above. The flow cytometer analysis of the vesicles showed a remarkable increase in EGFP and MAGEA4 specific signals while there was only a marginal difference between the control samples of non-incubated vesicles compared to vesicles incubated with EFGP (Figure 5E). These results are very similar to those which were obtained using vesicles isolated from EGFP and MAGEA4-EGFP expressing cells, confirming once more that MAGEA4 is able to direct recombinant proteins to EVs. Thus, the MAGEA4 protein can be used to load EVs with proteins that would not be packed into vesicles under normal cellular conditions.

## 3. Discussion

Extracellular vesicles possess a great potential to overcome medical challenges that have not yet been defeated by current techniques. EVs are shown to be promising candidates for fighting autoimmune diseases [59,60], organ injuries [61,62], virus resistance [63,64], cancer [65,66,67], and many more. However, before developing any EV-based therapy, it is crucial to know how to handle EVs to maintain their integrity and biological functions. In the current study, we studied the physiochemical properties of MAGEA4-EVs with a specific focus on the MAGEA4 protein. 

Our data show that MAGEA4-EVs are stable under common storage conditions as keeping at 4 °C and −80 °C for at least 3 weeks. Storing of EVs at −80 °C in phosphate buffer is proven to be the most trustworthy method and most commonly practiced [68]. Alternatively, keeping the vesicles at 4 °C for a short period may be more preferable as freezing and thawing may have damaging effects [69,70]. Although these conditions are considered to be the best, the number of vesicles have still been shown to decrease in either 4 °C and −80 °C [71,72]. Maroto et al. [55] detected an increase in the size of the vesicles in both conditions, while Sokolova et al. [73] reported a decrease if stored at 4 °C. Depending on the cargo, there are reports of maintenance or loss of cargo or both [74,75,76,77]. In our experiments, no statistically significant changes were detected at any temperature up to 3 weeks. 

Freezing and thawing of MAGEA4-EVs resulted in a steady decrease in the amount of MAGEA4 after each cycle, which turned to be significant by the end of the third cycle. This is consistent with reports evaluating freezing effects [55,69], however, some EVs seem to be more freeze-resistant [73,78,79]. We also observed swelling of vesicles after the first cycle, following a steady decline in the diameter after each subsequent cycle. It has been previously discussed that vesicles with lipid bilayer may become multilamellar due to forming of ice crystals and thus induce swelling [55]. On the other hand, if aggregates of EVs were formed due to strong forces of ultracentrifugation during isolation [80], then these could decompose due to freezing effects resulting in smaller particles detected and a more homogeneous population of EVs, which were both observed. Therefore, we speculate that the swelling and following decrease of vesicle diameter might be an outcome of the interplay between ultracentrifugation and freezing.

According to our working hypothesis, the MAGEA4 protein associates with EVs as a peripheral membrane protein. MAGEA4-EVs are resistant to the treatment with high salt but sensitive to high pH and non-ionic detergent. Incubation of vesicles with NaOH at pH 11.5 was deemed to be too harsh a treatment as the number of vesicles, along with the level of MAGEA4, dramatically decreased. This correlates with an earlier study showing the detrimental effect of high pH on EVs and their content [81]. Our data suggest that MAGEA4 binding to the surface of EVs is rather based on hydrophobic than electrostatic forces.

The MAGEA4 protein is able to associate with EVs as well as MLV Gag-induced VLPs in vitro. VLPs serve as an adequate model for EV research [53,82] as they bud from the cellular membrane similar to endogenous EVs but are more homogeneous in nature and can be produced in relatively high quantities [83]. Rather surprisingly, we discovered that MAGEA4 was able to bind to both vesicles as a result of robust passive incubation. If the purpose is to encapsulate the cargo, then loading of EVs with proteins is performed through active methods that induce transient permeability of the EVs such as electroporation, freeze-thawing, saponin treatment, and others [84,85,86]. Similar passive direct loading has been performed with apotransferrin to EVs with Tf receptor [87] or CD63 binding peptide [88]. Furthermore, MAGEA4 can be used to decorate the vesicles with recombinant proteins. Corso et al. [89] showed that a variety of EV markers with different localization are able to label EVs with different efficiencies, whereas transmembrane tetraspanins performed the best and soluble proteins the worst. Considering that MAGEA4 is a membrane-associated protein [53,54], it is able to display the recombinant protein on the surface of EVs noticeably well. This characteristic has the potential to be adapted for further EV-based application including labeling EVs for in vivo biological studies or loading EVs with recombinant proteins or peptides. However, more research lies ahead in order to fully discover the binding mechanism of MAGEA4 and to uncover its potential and efficiency to load EVs.

Extracellular vesicles have a great potential to be used as a tool for innovative therapeutic approaches. However, the natural vesicles still have some limitations, including low targeting capability and low concentration of functional molecules. The need for new and more accurate therapeutics and vaccines has led to the emergence of a new field called engineering of EVs [90]. Designing EVs is one way to overcome the limitations of natural EVs. For instance, anti-cancer EVs must specifically target cancer cells and transfer therapeutics such as drugs or short RNAs to tumor tissue without damaging the surrounding cells. Designer EVs will benefit simultaneously from their own functional molecules and newly loaded molecules and, combined with the antibody-based technologies, hold much promise to fight still untreatable diseases.

## 4. Materials and Methods

### 4.1. Cells and Plasmids

Transfection of mammalian cells was conducted as described in [53,54]. Mouse fibroblast cells COP5-ENBA were cultured in IMDM medium supplemented with 10% fetal calf serum, penicillin (100 U/mL), and streptomycin (100 ng/mL) at 37 °C. The cells were transfected with expression plasmids pQM-MAGEA4, pQM-MAGEA4-EGFP, pEGFP-C1, or pQMCF-MLV gag mixed with 50 μg of salmon sperm carrier DNA, then cultured in IMDM medium supplemented with 5% exosome free fetal calf serum, penicillin (100 U/mL), and streptomycin (100 ng/mL). MAGEA4 sequence, in pQM-MAGEA4 plasmid, is fused in-frame with C-terminal E2Tag epitope under the control of CMV promoter [53]. As a negative control, salmon sperm carrier DNA was used. Transfection was carried out at 230 V and 975 μF on the GenePulser Xcell™ (Bio-Rad Laboratories; Hercules, CA, USA). 

For protein production in bacteria, *E. coli* cells of BL-CodonPlus™ RP (Invitrogen; Waltham, MA, USA) strain was transformed with expression plasmids pET28a-EGFP, pET28a-MAGEA4 and pET28a-MAGEA4-EGFP encoding respective recombinant proteins with His-tag in N-terminus. For generation of pET28a-MAGEA4-EGFP the EGFP encoding sequence was cloned to the C-terminus of MAGEA4 encoding sequence from pQM-MAGEA4-EGFP. 

### 4.2. Proteins

The protein production and purification were performed as previously [53]. The transformed bacteria were grown at 37 °C to the optical density of 0.6 measured at 600 nm using Ultraspec 7000 spectrophotometer (GE Healthcare Life Sciences; Marlborough, MA USA) following induction of protein expression with 1 mM IPTG for 2 h at 37 °C for EFGP and MAGEA4 or 24 °C for MAGEA4-EGFP. Then the cells were collected by centrifugation at 5000× *g* for 15 min at 4 °C using Centrifuge 5810R (Eppendorf; Hamburg, Germany) and resuspended in the buffer containing 50 mM Tris pH 8.0 and 500 mM NaCl. Proteins were purified with Ni-Sepharose™ 6 Fast Flow beads (GE Healthcare Life Sciences; Marlborough, MA USA) under standard native conditions following the manufacturer’s recommendations; 20 mM imidazole was added to the buffer for binding reactions, 40 mM for wash buffers and 250 mM for elution of proteins from the beads. After purification, the buffer was exchanged to PBS with Amicon^®^ Ultra centrifugal filters (Merck; St. Louis, MO, USA) and the concentration of proteins was determined by the Bradford protein assay (Bio-Rad Laboratories; Hercules, CA, USA) using bovine serum albumin as a standard. 

### 4.3. Isolation and Purification of EVs and VLPs

Isolation of vesicles was carried out as described in [53,54]. Usually, the purification was performed from 35 mL of cell culture media obtained from 5.4 × 10^6^ cells transfected with 2.5 μg of MAGEA4 expression plasmid. The media was collected 72 h after transfection and centrifuged as described. The first centrifugation at 300× *g* for 10 min was carried out to remove dead cells and cell debris.

For isolation of EVs, the supernatant was centrifuged at 2000× *g* 20 min 4 °C to precipitate apoptotic bodies and other vesicles of similar size. The next centrifugation was carried out at 16,500× *g* for 20 min 4 °C and the third at 120,000× *g* using Optima™ L-90K Ultracentrifuge with rotor SW28 (Beckman Coulter; Brea, CA, USA) for 70 min at 4 °C to precipitate small EVs. The small EV pellet was suspended in 200 μL of PBS. Washing the vesicles with PBS included centrifuging EVs at 120,000× *g* for 90 min at 4 °C using the Optima™ L-90K Ultracentrifuge with rotor SW55Ti. The final EVs were resuspended in 200 μL of Dulbecco’s PBS (DPBS) (Merck; St. Louis, MO, USA). The EV sample concentrations were measured with the Bradford Protein Assay (Bio-Rad Laboratories; Hercules, CA, USA) using BSA as a standard. The average protein concentration of EVs obtained from 35 mL of cell culture media was 1.8 mg/mL and the amount of particles 4.9 × 10^11^.

For isolation of VLPs after the initial centrifugation at 300× *g* the supernatant was further centrifuged at 120,000× *g* using Optima™ L-90K Ultracentrifuge with rotor SW28 for 3 h at 4 °C through 5 mL of 20% sucrose cushion in PBS. Washing, resuspension in PBS, and concentration measurement were performed identically to EV samples described above. 

### 4.4. Physico-Chemical Treatment of MAGEA4-EVs

For storage, freeze-thaw, and chemical treatment experiments purified MAGEA4 carrying EVs were divided equally into aliquots, including 30 μg of EVs as determined by the Bradford assay. The volume of the aliquots was increased to 100 μL with DPBS. After treatment, each EV aliquot was used for the Western blot, flow cytometry, and NTA analyses. All the treatments were performed in at least 3 replicates. 

For the storage experiment, an aliquot without any further treatment was used as the control sample, while 3 aliquots were used for incubation at 4 °C and another 3 for −80 °C up to 21 days. After each 7 days, an aliquot from both treatments was removed for the analyses. Similarly, the freeze-thaw experiment had an aliquot of the EVs without any further treatment as the control sample. However, the treatment involved subjecting 3 aliquots up to three freeze-thaw cycles comprising of 1 h of freezing at −20 °C and 20 min of thawing at room temperature. After each cycle, an aliquot was removed and kept at 4 °C until analyzed. The chemical treatment involved suspending the EV aliquots in 1 mL of the following PBS-based solutions: 1 M NaCl, 0.33 M MgCl_2_, 10 mM EDTA, 10 mM EGTA, 3.5 mM NaOH with pH 11.5, 0.02% Triton X-100 and pure PBS as the control. The samples were incubated for 1 h at room temperature on the bench and washed through UC. The pellet was resuspended in PBS. 

### 4.5. In Vitro Binding Experiment

EVs and VLPs, purified from cell culture media of COP5-EBNA cells without the introduction of MAGEA4 expression plasmid, were incubated with purified MAGEA4 protein for 1 h at room temperature. For this, 20 μg of the vesicles were suspended with 10 μg of the protein solution, and the volume was increased to 50 μL with PBS. After incubation, the vesicles were washed through UC as described above. The pellet was resuspended in 100 μL PBS. Alternatively, VLPs were also purified through SEC. After incubation, as 100 μL was the recommended minimal sample volume for SEC column (HansaBioMed Life Sciences; Tallinn, Estonia), 50 μL PBS was added, following loading of the sample to the SEC column. The VLP sample was fractionated into 26 aliquots of 100 μL, additional PBS was loaded to the column as required. Fractions 4–24 were used for Western blotting analysis.

For in vitro EGFP loading experiment, 30 μg of EVs were incubated with purified 20 μg of EGFP or 40 μg of MAGEA4-EGFP chimeric protein for 1 h at room temperature. The incubation volume 100 μL was obtained by adding PBS. After incubation, the vesicles were washed through UC and the pellet was resuspended in PBS.

### 4.6. Western Blot Analysis

Cellular, vesicular, or purified protein samples were suspended in Laemmli buffer and denatured for 10 min at 100°. The lysates were separated electrophoretically using 10% SDS-PAGE gel and blotted onto a PVDF membrane using Trans-Blot SD Semi-Dry Transfer Cell (Bio-Rad Laboratories; Hercules, CA, USA). Affinity-purified rabbit polyclonal antibodies against MAGEA4 (2.5 mg/mL) [53] were used for immunoblotting at dilutions of 1:10,000. Alpha-tubulin (dilution 1:4000; T5168; Merck; St. Louis, MO, USA), anti- TSG101 (dilution 1:10,000, T5701; Merck; St. Louis, MO, USA), anti-E2Tag antibody 5E11 (dilution 1:10,000; Icosagen; Tartu, Estonia) were used in different experiments. Goat anti-rabbit (1 mg/mL, LabAS; Tartu, Estonia) and goat anti-mouse (1 mg/mL, LabAS) antibodies conjugated with HRP were used as secondary antibodies at a dilution of 1:10,000. Protein signals were detected using ECL Western blotting (GE Healthcare; Marlborough, MA USA) reagents. The staining of SDS-PAGE gels was performed with PageBlue Protein Staining Solution (Thermo Scientific; Waltham, MA, USA). 

### 4.7. Flow Cytometry

Flow cytometry analyses were conducted to analyze the expression of EGFP in live cells and vesicles, also the surface expression of MAGEA4 protein in vesicles. For live-cell analysis, COP5-EBNA cells were collected 72 h post-transfection and suspended in 1 mL PBS. Cells were then washed by centrifugation and resuspended in PBS, and analyzed with Attune NxT Flow Cytometer (Thermo Fischer Scientific; Waltham, MA, USA) using the official software for the apparatus. Statistical analysis and data graphical representation of all flow cytometry experiments were carried out with the FlowJo VX software 7.6.5 (Becton, Dickinson and Company; Franklin Lakes, NJ, USA).

Analyses of vesicles were performed using a bead-assisted protocol. 20 μg of EVs from each sample were incubated with 10 μL of 4 μm diameter aldehyde/sulphate latex beads (Invitrogen; Waltham, MA, USA) for 15 min at room temperature. Then PBS was added to a final volume of 1 mL, and the mixture was incubated at 4 °C overnight using end-over-end rotation. The beads were then blocked with a 100 mM glycine/PBS solution for 30 min at 4 °C and washed twice with 2% BSA in PBS. Incubation with affinity-purified rabbit polyclonal antibodies against MAGEA4 (final concentration of 1 ng/μL) was carried out in 2% BSA in PBS for 1 h at 4 °C using end-over-end rotation. The samples were then washed twice. Anti-rabbit or anti-mouse Alexa 488 or Alexa 568 antibodies (1 mg/mL, dilution 1:1000, Invitrogen) were used as secondary antibodies in incubations for 1 h at 4° C using end-over-end rotation. The beads were washed twice, resuspended in 500 μL of 2% BSA in PBS, and analyzed with Attune NxT Flow Cytometer (Thermo Fischer Scientific; Waltham, MA, USA).

### 4.8. Analysis of EVs by NTA

NTA (Nanoparticle tracking analysis) was performed with ZetaView nanoparticle analyzer (Particle Metrix GmbH; Inning am Ammersee, Germany). Before each session, the machine was calibrated using 102 nm polystyrene beads. In all cases, 11 measurements were recorded twice and averaged in at least 1 dilution in DPBS and analyzed using the ZetaView Software 8.04.02 (Particle Metrix GmbH) using default image evaluation settings and following camera acquisition settings: sensitivity 85, shutter 70, and frame rate 30.

### 4.9. Statistical Analysis

FlowJo VX (Flow Jo LLC) was used for creating histograms and dot plots of flow cytometer experiments. Prism 8.4.3. (GraphPad, San Diego, CA, USA) was used to conduct statistical analysis and draw diagrams of the corresponding data. For calculation of *p*-values in the storage experiment, control sample values were compared to different time points in the groups using 2-way Anova with Bonferroni’s multiple comparisons test. The *p*-values of the freeze-thaw experiment were calculated using the one-way ANOVA with Dunnett’s multiple comparison test, and in the chemical treatment experiment, the one-sample t-test was implemented.

## Figures and Tables

**Figure 1 ijms-22-05208-f001:**
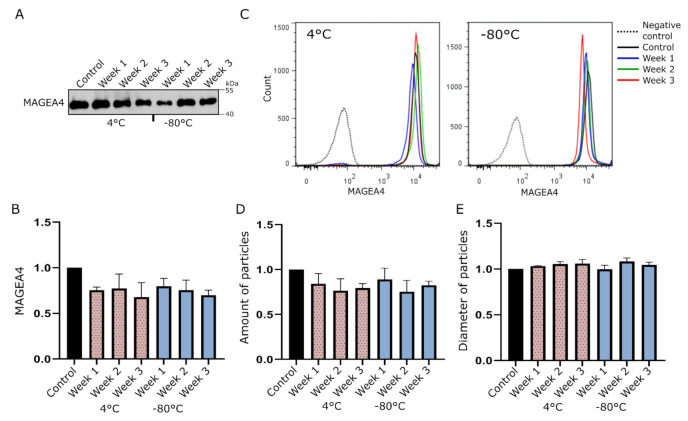
MAGEA4 carrying extracellular vesicles (MAGEA4-EVs) are stable under common EV storage conditions. MAGEA4-EVs were incubated at −80 °C or 4 °C for up to 3 weeks, and a sample from each group was analyzed after every 7 days. (**A**) Western blot analysis of MAGEA4-EVs at different time points with antibodies against the MAGEA4 protein. (**B**,**C**) The relative surface binding of MAGEA4 analyzed by flow cytometry using anti-MAGEA4 and Alexa 488-conjugated secondary antibodies. The relative mean fluorescence intensity (MFI) values (**B**) of an average of three experiments and fluorescence profiles (**C**) of a representative experiment are shown. (**D**) The amount and (**E**) average diameter of vesicles as measured by the nanoparticle tracking analysis (NTA). Data shown are average of three independent experiments. In all cases, the relative values, where the control was set as 1, are shown. Freshly prepared MAGEA4-EVs serve as a negative control.

**Figure 2 ijms-22-05208-f002:**
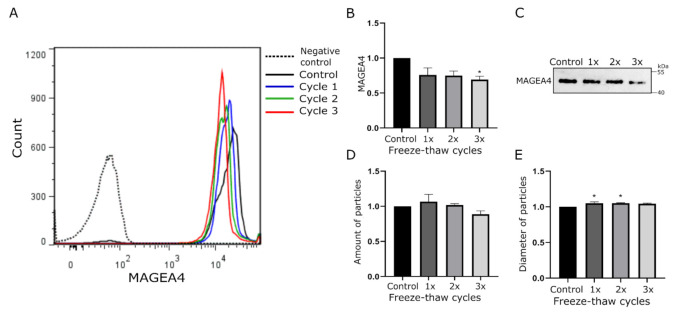
The effect of freezing-thawing cycles on MAGEA4-EVs. MAGEA4-EVs were subjected up to three freeze-thaw cycles. A single cycle comprised of 1 h of freezing at −20 °C and 20 min of thawing at room temperature. (**A**) Analysis of MAGEA4 on the surface of EVs using flow cytometry with anti-MAGEA4 and Alexa 488-conjugated secondary antibodies. (**B**) The relative amount of MAGEA4 on the surface of EVs (MFI) analyzed by flow cytometry. (**C**) Western blot analysis of samples with anti-MAGEA4 antibodies. (**D**) The amount and (**E**) average diameter of vesicles as measured by the NTA. Data shown are average of three independent experiments. In all cases, the relative values where the control was set as 1 is shown. *—The difference of the sample value compared to the control sample value is statistically significant with *p*-value <0.05.

**Figure 3 ijms-22-05208-f003:**
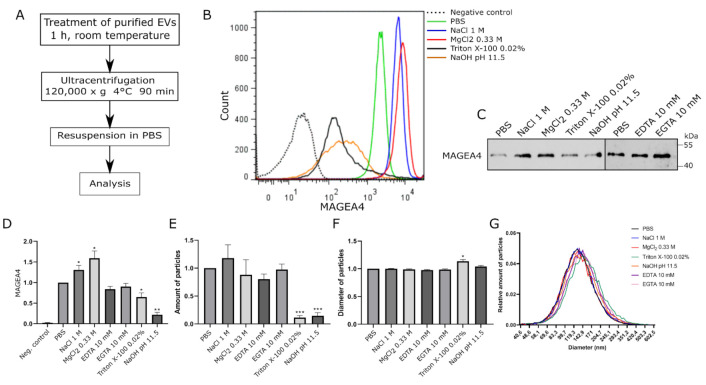
MAGEA4-EVs are resistant to high salt concentrations but vulnerable to high pH. (**A**) The scheme of the experiment. MAGEA4-EVs were treated with different chemicals for 1 h at room temperature. After the treatment, vesicles were purified by ultracentrifugation through PBS at 120,000× *g* for 90 min. (**B**) Analysis of MAGEA4 on the surface of EVs using flow cytometry with anti-MAGEA4 and Alexa 488-conjugated secondary antibodies. The dotted line is a negative control (latex beads without EVs), and PBS marks the positive control (MAGEA4-EVs without any treatment). (**C**) Western blot analysis of MAGEA4-EVs after treatment with chemicals using antibodies against the MAGEA4 protein. (**D**) The mean fluorescence intensity (MFI) values of the corresponding flow cytometry histograms in (**B**). (**E**) The amount and (**F**) average diameter of vesicles as measured by NTA. Data shown are average of three independent experiments. In all cases, the relative values, where the control was set as 1, are shown. (**G**) The relative amount of EVs by diameter as measured by NTA. The results indicated with * had a statistically significant difference compared to value of PBS-treated sample: * *p*-value < 0.05; ** *p*-value < 0.01; *** *p*-value < 0.001.

**Figure 4 ijms-22-05208-f004:**
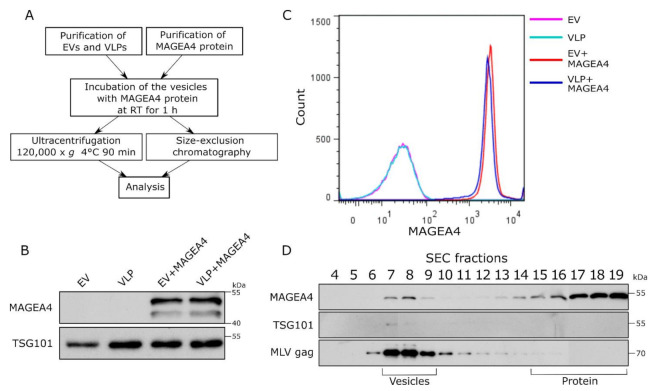
MAGEA4 is able to bind to the surface of vesicles in vitro. (**A**) The scheme of the experiment. Purified EVs and VLPs were incubated with the MAGEA4 protein for 1 h at room temperature following purification through ultracentrifugation or fractionation by SEC. (**B**) Western blot analysis of EVs after incubation with the MAGEA4 protein and purification by ultracentrifugation using antibodies against the MAGEA4 protein. (**C**) Analysis of MAGEA4 binding to the surface of EVs after in vitro binding and purification by ultracentrifugation. Flow cytometry was performed with anti-MAGEA4 and Alexa 488-conjugated secondary antibodies. (**D**) Western blot analysis of SEC fractions after the incubation of MAGEA4 and VLPs in vitro. Antibodies against MAGEA4, TSG101, and MLV Gag were used. The positions of vesicles and proteins according to the manufacturer are shown.

**Figure 5 ijms-22-05208-f005:**
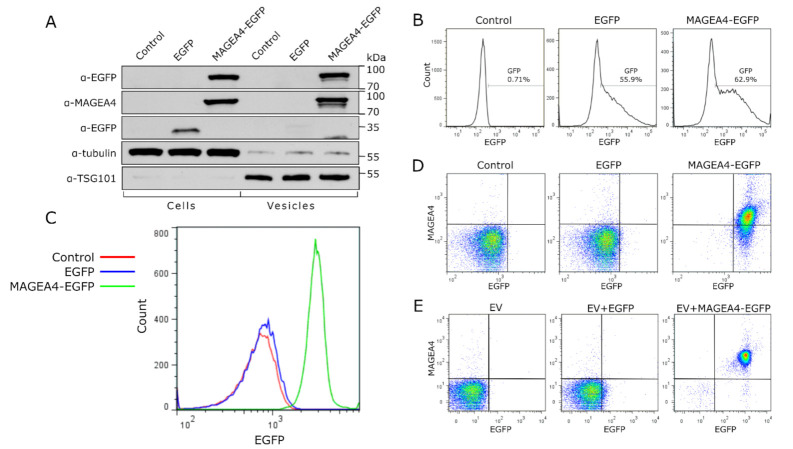
MAGEA4 is able to load the marker protein EGFP onto EVs. (**A**) Western blot analysis of cells transfected with expression plasmid for MAGEA4-EGFP and EGFP proteins and EVs isolated from the cell culture media of transfected cells. Control—mock cells transfected with empty vector. (**B**) The transfection efficiency of cells transfected with expression plasmid for MAGEA4-EGFP and EGFP. (**C**) The EGFP fluorescence profile of EVs isolated from the cell culture media of transfected cells. Control—mock cells transfected with empty vector. (**D**) The dot blot of EVs isolated from the cell culture media of transfected cells. EV-bound latex beads were incubated with anti-MAGEA4 and Alexa 568-conjugated secondary antibodies. (**E**) Dot blot analysis of in vitro binding experiment. EVs from non-transfected cells were incubated with purified EGFP or MAGEA4-EGFP protein or with no protein as control and purified through ultracentrifugation. Signals of EV-bound latex beads obtained with anti-MAGEA4 and Alexa 568-conjugated secondary antibody (for MAGEA4) and the fluorescence of EGFP are shown.

## Data Availability

The data presented in this study are available on request from the corresponding author.

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
