# Peer review of "MAGEA4 Coated Extracellular Vesicles Are Stable and Can Be Assembled In Vitro"

_ijms, 2021, doi:10.3390/ijms22105208_

Round 1
Reviewer 1 Report
The authors provide interesting results about the resistance of MAGEA4-EVs to several stresses, including freezing-thaw cycles, high pH, and high salt. They also explore the ability of MAGEA4 protein to bind to EVs. However, several aspects could have been investigated more.
Specifically, the following points are mandatory for publication:
- a full characterization of EVs is missing. According to ISEV guidelines (MISEV2018: https://doi.org/10.1080/20013078.2018.1535750), it would be appropriate to characterize the EVs of the study by describing in a dedicated figure: (a) volume of fluid, and/or cell number, used to isolate EVs, (b) protein amount and particle number, (c) presence of transmembrane or GPI-anchored protein localized in cells at plasma membrane or endosomes, cytosolic protein with membrane-binding or -association capacity, presence/absence of expected contaminants (At least one each of the three categories above), (d) images of single EVs by wide-field and close-up (e.g. electron microscopy, scanning probe microscopy, superresolution fluorescence microscopy).
- in order to confirm EV integrity after every treatment analyzed in the study, images of single EVs (by one of the techniques listed above) should be provided.
- as required by journal guidelines, original images for blots must be available for reviewers and plasmids sequences must be reported.
Moreover, the following points should be addressed to improve the manuscript for publication:
- it would be interesting to analyze the variation in the size distribution of EVs after different treatments, non only by the mean, but also analyzing mode and distributions D10, D50, and D90.
- regarding EV stability to pH, it would be interesting to analyze the stability after treatment with increasing pH and at least another basic substance.
Author Response
Reviewer #1
- We have added a full characterization of MAGEA4-EVs to the manuscript. We made a dedicated figure (Suppl. Fig.S1), where the scheme of purification, volume of fluid used for purification, protein amount and particle number are shown together with western blot analysis and TEM image of purified vesicles. We also added this information to Materials and Methods section (lines 372-374 and 385-387).
- The reviewer asked us to confirm EV integrity after every treatment… Unfortunately, we were not able to address this question. All our experiments are made with freshly prepared EVs and we were not able to make them with such a short time. The only left-overs we could find were used to make additional experiment for point 5. We do not have any clue that the treatment with salt or EDTA/EGTA somehow damages the EVs. But the question is justified in the case of treatment with high pH and detergent. A drastic drop in the amount of EVs was seen after the treatment with NaOH pH=11.5 which shows that they were probably damaged. However, our EV fraction consists of different EV sub-populations which may differently respond to the high pH. As shown on Fig. 3B, some MAGEA4 protein is still detected after the treatment. MAGEA4-EVs are not classical exosomes and do not contain any tetraspanins so it is very difficult to follow them with some other specific marker except for its own antibodies.
- We have added all original images and plasmid sequences to the submission.
- The data of size distribution (mode and distributions D10, D50 and D90) are added as supplementary Fig.S2 to the manuscript.
- We made an additional experiment with another basic substance – KOH at the same pH, pH=11.5. The data of this experiment are added to the manuscript as Supplementay Fig. S3. Again, treatment with high pH was detrimental to MAGEA4-EVs.
We agree with the reviewer that experiments with increasing pH will be very interesting. In this case, it will be also interesting to analyze the effect of high pH on exosomes (using CD63 or CD9 as a marker) and other EVs in addition to MAGEA4-EVs.
Reviewer 2 Report
The manuscript reports the results of a study aimed at characterizing biochemical properties of MAGEA4-coated extracellular vesicles. The presented study further develops observations previously obtained by the authors on the same topic. Features of this type of vesicles such as storage conditions, changes following exposure to salts, chelators and detergents, and interactions with other possible vesicle cargos have been analyzed. In view of the increasing interest on extracellular vesicles as potential targets or tools for therapeutical intervention the study adds substantial new technological information in this field.
Some minor points can be suggested to further improve the paper:
Line45: in addition to the very recent mentioned papers, some of the first evidence on the role of EV in cardiovascular functions (e.g. Exp Cell Res 316:1977-1984, 2010) can be provided for completeness.
Section 2.3 (line 141-193): The section addresses different types of treatment. To allow an easier reading, it could be structured in sub-sections each illustrating the results of a specific type of treatment.
Discussion: This section is quite extensive (lines 256-367) and probably it can be reduced to some extent. Biological/therapeutic issues related to the obtained findings, however, are poorly addressed, being limited to the last 3 lines. Since in their previous study (Oncotarget 10:3694-3708, 2019) the authors emphasized the role of MAGEA-EV as a mechanism potentially inducing cancer formation, while in the present discussion the possible use in anti-cancer applications is mentioned, these aspects would deserve some more comment.
Author Response
Reviewer #2
- Thank you for paying our attention to this early publication. We have added it as ref 20 to the revised manuscript (line 46).
- Section 2.3: We have re-structured this section (lines 142-195). Hopefully it is easier to follow now.
- Discussion: We have re-written this section to make it more compact and better focused. Now it takes lines 259-336. We added one paper (ref. 90) which fits this topic.